# Enhancing Open-Vocabulary Object Detection through Multi-Level Fine-Grained Visual-Language Alignment

## Abstract

Traditional object detection systems are typically constrained to predefined categories, limiting their applicability in dynamic environments. In contrast, open-vocabulary object detection (OVD) enables the identification of objects from novel classes not present in the training set. Recent advances in visual-language modeling have led to significant progress of OVD. However, prior works face challenges in either adapting the single-scale image backbone from CLIP to the detection framework or ensuring robust visual-language alignment. We propose Visual-Language Detection (VLDet), a novel framework that revamps feature pyramid for fine-grained visual-language alignment, leading to improved OVD performance. With the VL-PUB module, VLDet effectively exploits the visual-language knowledge from CLIP and adapts the backbone for object detection through feature pyramid. In addition, we introduce the SigRPN block, which incorporates a sigmoid-based anchor-text contrastive alignment loss to improve detection of novel categories. Through extensive experiments, our approach achieves 58.7 AP for novel classes on COCO2017 and 24.8 AP on LVIS, surpassing all state-of-the-art methods and achieving significant improvements of 27.6% and 6.9%, respectively. Furthermore, VLDet also demonstrates superior zero-shot performance on closed-set object detection.

## 1 Introduction

With the development of visual-language models (VLM) (Radford et al., 2021; Zeng et al., 2023; Zhai et al., 2023), recent approaches have extended traditional object detection frameworks into open-world object detectors capable of detecting objects from categories not present in the training set (Li et al., 2022a; Zhong et al., 2022; Liu et al., 2023; Cheng et al., 2024; Zhou et al., 2022; Gao et al., 2022). As a pioneer, OVR-CNN (Zareian et al., 2021) proposed open-vocabulary object detection by introducing a pre-training stage that uses image-caption pairs to align image and text encoders. Then, the detection model employs the aligned image encoder as its backbone, classifying bounding boxes based on the similarity between image features and text embeddings of category names generated by the text encoder. Although subsequent works (Gu et al., 2021; Wu et al., 2023a) have improved the performance by distilling knowledge from CLIP (Radford et al., 2021) for region-wise classification, they still rely on visual-language knowledge aligned on image-level. A fine-grained region-wise visual-language alignment is desired for more accurate detection (Zhong et al., 2022). Many works with region-wise alignment have been proposed to bridge this gap, such as RegionCLIP (Zhong et al., 2022) and YOLO-World (Cheng et al., 2024). However, these approaches also bring several challenges. First, they struggle to adapt image encoders from CLIP into detection models. Some methods simply adopt the single-scale backbone from CLIP, which compromises spatial information and detection accuracy. Others attempt to train a multi-scale backbone from scratch using pseudo-labeled regions (*i.e.* bounding boxes) generated by zero-shot detectors, such as GLIP (Li et al., 2022a), on large-scale datasets. These pseudo labels demand significant additional processing efforts (Li et al., 2022a; Yao et al., 2022; Liu et al., 2023; Cheng et al., 2024), which greatly limits their reproducibility. Second, these methods bias toward region-wise alignment, ignoring that image-wise alignment is also beneficial for visual-language modeling (Zeng et al., 2023). Finally, aforementioned methods either adopt a single-stage detector, or leverage a vanilla Region Proposal

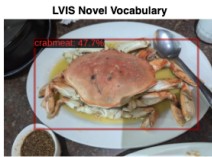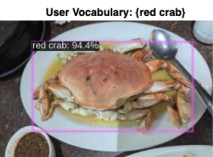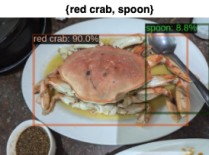

Figure 1: With multi-level fine-grained visual-language alignment, VLDet accurately detects objects from LVIS categories to user-specified ones.

Network (RPN). A sophisticated RPN can significantly boost OVD performance by proposing objects of any potential classes (Gu et al., 2021).

To tackle these challenges, we propose a novel framework, Visual-Language Detection (VLDet). By introducing a Visual-Language Pyramid Upscale Block (VL-PUB), we integrate CLIP's image and text encoders into our framework to effectively exploit its enriched visual-language latent space and construct a feature pyramid to detect objects of varying sizes. To enhance the visual-language alignment, we incorporate contrastive losses at multiple levels, achieving multi-level fine-grained alignment. Specifically, we employ: (i) an image-caption contrastive loss, which provides a broader understanding of the relationship between full images and captions. This is conducted within mini-batches to balance training between image-wise and region-wise visual-language alignment; (ii) a region-text contrastive loss (Li et al., 2022a) to achieve fine-grained alignment between image regions and object category texts; (iii) an additional anchor-text binary visual-language alignment, which enhances the RPN ability to differentiate background from objects of any category and facilitate learning general semantic information applicable to any category, thereby improving generalizability to novel categories, as illustrated in Fig. 1. Notably, VLDet is pre-trained solely on the detection dataset, Objects365 (Shao et al., 2019), where arbitrary foundation VLM can be leveraged to generate captions. This approach offers significant efficiency and reproducibility compared to the pseudo-labeling of bounding boxes for large-scale image-caption pairs. Through comprehensive experiments on public datasets for OVD, VLDet outperforms all baselines and also demonstrates extraordinary performance of zero-shot inference and fine-tuning for traditional object detection.

To summarize, our main contributions are as follows,

- We propose a VL-PUB module to effectively adapt image and text encoders from CLIP into open-world object detector to leverage the enriched visual-language latent space and utilize multi-scale image features for enhanced detection performance.
- We design SigRPN, the first RPN for OVD that uses sigmoid-based visual-language contrastive learning to differentiate between background and objects of any class, aiding in proposing objects of all potential new categories and significantly enhancing generalization to novel categories in the OVD scenario. We further integrate image-wise contrastive loss for more robust visual-language alignment.
- Through extensive experiments on public datasets, the proposed VLDet outperforms state-of-the-art methods in OVD, particularly enhancing the mAP of novel categories for OVD from 46.0 to 58.7 on COCO2017, and from 23.2 to 24.8 on LVIS, showing the great generalization capability of our VLDet.

## 2 RELATED WORK

### 2.1 TRADITIONAL OBJECT DETECTION

Traditional object detectors are trained on datasets with predefined categories and subsequently detect objects within this fixed set of categories (*i.e.*Closed-set). Popular object detectors can be categorized into three groups: single-stage, two-stage, and query-based. YOLOs (Redmon, 2016; Redmon & Farhadi, 2017; Redmon, 2018; Jocher & Qiu, 2024) are the representative works from the first category, which utilize convolutional architectures for real-time object detection. DETR (Carion et al., 2020) pioneered the use of transformers for object detection, inspiring numerous query-based methods (Kamath et al., 2021; Zhang et al., 2022; Zong et al., 2023; Ren et al., 2023). Two-stage detectors (Ren et al., 2016; He et al., 2017; Cai & Vasconcelos, 2019), such as Faster R-CNN (Ren

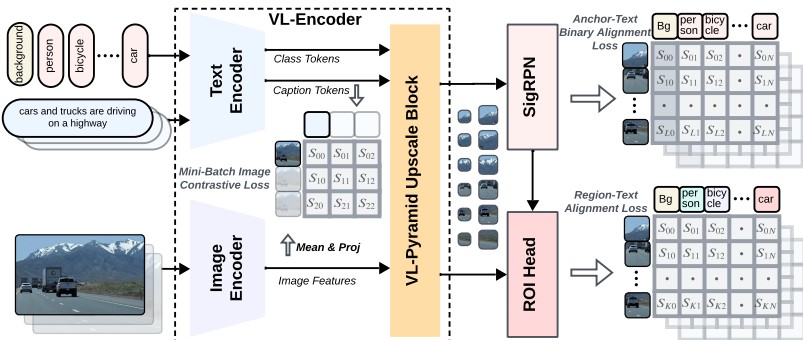

Figure 2: With multi-level visual-language alignment, VLDet is a unified network which can effectively exploit enriched visual-language semantic space from pre-aligned single-scale backbones while also providing pyramid image features for improved performance in OVD.

et al., 2016) employ a two-stage framework for proposal generation and Region-of-Interest (RoI) classification and regression. While RPN in two-stage learns general information to distinguish background from objects across all categories, whether novel or base, this paper emphasizes the importance of RPN for OVD.

## 2.2 OPEN-WORLD OBJECT DETECTION

There are two primary branches in the realm of open-set object detection: open-vocabulary object detection (OVD) and zero-shot object detection (ZOD).

OVD was first proposed by OVR-CNN (Zareian et al., 2021), where a visual encoder is pre-trained on image-text pairs to learn object concepts, enabling the detection of additional categories not included in the training set. Subsequently, ViLD (Gu et al., 2021) and DetPro (Du et al., 2022) trained object detectors by distilling visual features from a pre-trained CLIP model. RegionCLIP (Zhong et al., 2022) further advanced this by learning region representations from "pseudo" region-text pairs provided by a pre-trained CLIP model. Detic (Zhou et al., 2022) trained the classifier of a detector on ImageNet21K with broader vocabularies. BARON (Bica et al., 2024) groups contextually interrelated regions as a bag and aligns the embedding of the bag of regions beyond individual regions.

In the context of ZOD, recent methodologies, beginning with GLIP (Li et al., 2022a; Liu et al., 2023; Shen et al., 2024; Cheng et al., 2024; Jiang et al., 2024), achieve it by integrating the phrase grounding task with object detection. This integration leverages grounding datasets (*e.g.*GoldG curated by (Kamath et al., 2021) from (Plummer et al., 2015; Krishna et al., 2017; Hudson & Manning, 2019)) and the large-scale image-caption pairs (Sharma et al., 2018; Ordonez et al., 2011) with pseudo-labeled bounding boxes, which offer a multitude of categories by extracting words from image captions. Our approach primarily targets OVD, yet it also demonstrates promising ZOD capability, achieved by pre-training solely on Objects365 and avoiding bounding box pseudo-labeling, thereby enhancing reproducibility.

## 3 APPROACH

Single-scale ViT backbones are widely adopted by researchers for visual-language alignment with image-caption pairs (Radford et al., 2021; Zareian et al., 2021; Zhai et al., 2023). They produce enriched multi-modal embedding space but yield limited spatial information. To address this, VLDet is designed to unify any single-scale visual-language backbones into object detectors, providing features at various scales for enhanced detection, as illustrated in Fig. 2.

### 3.1 OVERALL ARCHITECTURE

To leverage the enriched semantic information from CLIP's aligned image and text encoders, we design a Visual-Language Encoder (VL-Encoder) that integrates these two encoders with a Visual-Language Pyramid Upscale Block (VL-PUB). While the CLIP image encoder extracts features at a single scale, VL-PUB casts the output features into multiple scales after fusing them with text

features, enhancing detection performance for objects of different sizes (Lin et al., 2017). Previous work (Li et al., 2022a) grouped all class names together and generated embeddings for this large sequence, which led to two significant issues: first, it assigns a different number of tokens to different classes, requiring token grouping when computing similarities with visual features; second, non-trivial adjustment is required when the dataset (*e.g.*Object365 or LVIS) contains too many classes, which exceeds the maximum token length of the text encoder (*e.g.*256); In contrast, we embed each category name into one single token using the CLIP text encoder, which is more systematic for contrastive loss computation and naturally supports an arbitrary number of categories without additional modifications. Moreover, it's worth noting that we only use class names as text input during inference, while an extra image caption branch is included at training stage. This design benefits text feature extraction for class names by providing more textual context during training. By leveraging such an informative text feature when conducting image-wise contrastive learning, we further enhance the visual-language alignment for object detection.

Specifically, an input image, $x \in \mathbb{R}^{H \times W \times 3}$ ($H$ is image height, and $W$ is the width), first goes through the image encoder and is embedded into a single-scale image feature, $v_0 \in \mathbb{R}^{\frac{H}{p} \times \frac{W}{p} \times C_v}$, where $C_v$ is the number of visual dimensions and $p$ is patch size. Class prompts are embedded into language feature, $l_{cls} \in \mathbb{R}^{N \times C_l}$, where $N$ is the number of classes in the training data including *background* class (*e.g.*366 for Objects 365) and $C_l$ is the dimensions of language feature. The image caption is also embedded by text encoder into a single token, $l_{cap} \in \mathbb{R}^{1 \times C_l}$. We add a linear layer to project the mean of extracted image feature $v_0$ over all visual tokens to the same dimension of caption embedding $l_{cap}$ for image-level visual-language contrastive learning, leading to better visual-language alignment.

To better capture the objects of various sizes, the extracted text features $l_{cls}, l_{cap}$ and image feature $v_0$ first go through VL-PUB to generate image features at multiple scales. This block first fuses the image feature with text feature via bi-directional cross-attention (Li et al., 2022a), then employ a single transformer layer to further embed text features into deeper embeddings $l'_{cls} \in \mathbb{R}^{N \times C_l}$ and leverage multiple Deconvolution, MaxPooling and Convolution layers to generate image feature at various scales $v_i$, where $i \in \{1, 2, ..., Z\}$ and $Z$ is a hyper-parameter, denoting the number of output scales. The scales of generated image features range from $\frac{4H}{p} \times \frac{4W}{p}$ to $\frac{H}{4p} \times \frac{W}{4p}$, and the visual dimension $C_v$ becomes 256. For simplicity, we use $C_v = 256$ in the following sections.

The extracted multi-scale image features $v_i, i \in \{1, 2, ..., Z\}$ and text feature $l'_{cls}$ are fed into an Visual-Language RPN module (SigRPN) for proposal prediction of any potential object classes. In SigRPN, the text feature is first fused with multi-scale image features through bi-directional cross-attention (Li et al., 2022a) then goes through a single transformer layer to get a more informative feature $l''_{cls} \in \mathbb{R}^{N \times C_l}$. After the fusion with text feature, the image features with various scales are fed into several convolution layers. Each scale $v_i \in \mathbb{R}^{h_i \times w_i \times C_v}$ ($h_i$ and $w_i$ denotes the height and width of the feature map for the $i$th scale) generates the corresponding *objectness logits* $a_i \in \mathbb{R}^{h_i \times w_i \times A \times C_l}$, where $A$ denotes the number of anchors for each spatial position and *anchor deltas* $d_i \in \mathbb{R}^{h_i \times w_i \times A \times 4}$ (Ren et al., 2016). Finally, based on the proposals from SigRPN, ROI Head extracts region-wise features from the multi-scale image features $v_i, i \in \{1, 2, ..., Z\}$ for object classification ($Z = 5$ in our experiment).

## 3.2 Visual-Language Pyramid Upscale Block

Although the pre-aligned image and text encoders from CLIP contain enriched semantic information, having been trained on a dataset with 400M image-caption pairs, the image encoder outputs visual features at only a single scale. Multi-scale image features typically perform much better (Lin et al., 2017; Li et al., 2022b) for object detection, as objects in images vary in size, and multi-scale feature maps can provide more comprehensive spatial information. Follwing the same idea, before feeding the image features generated by the image encoder directly into the subsequent RPN and ROI Head, we introduce a Visual-Language Pyramid Upscale Block (VL-PUB) to extract feature maps at various scales, as shown in Fig. 3.

Specifically, suppose we have obtained the text features $l_{cls}$, $l_{cap}$ and image feature $v_0$ from text encoder $E_L$ and image encoder $E_V$ as follows,

$$l_{cls} = E_L(\text{CLS}), \;\; l_{cap} = E_L(\text{CAP}), \;\; v_0 = E_V(x), \tag{1}$$

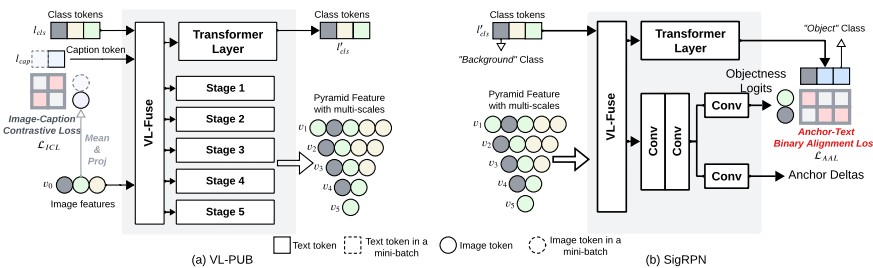

Figure 3: (a) $\mathcal{L}_{ICL}$ for image-wise visual-language alignment is computed before VL-PUB module. After the fusion of image feature and text feature, VL-PUB generates deeper text feature and pyramid image feature at various scales. (b) SigRPN computes $\mathcal{L}_{AAL}$ for fine-grained visual-language alignment on the general difference of *Background* and *Object* of any class.

where $x$ is the input image, CAP, CLS denotes the corresponding caption and class prompts. We encode CLS and CAP separately to reduce the memory consumption and computation by reducing the number of tokens for CLS since the category names are much shorter than captions.

Previous works proved that early fusion of visual and language features leads to enhanced performance (Li et al., 2022a; Liu et al., 2023). We first employed an visual-language fuse layer (VL-Fuse) to enhance both text and image features with bi-directional cross-attention (Detailed in Sec. A) after concatenating the class tokens and caption token as follows,

$$v_f, l_f = \text{VLFuse}(v_0, \text{Concat}(l_{cls}, l_{cap})) \tag{2}$$

where $v_f$, $l_f$ denotes the fused visual feature and language feature. Next, the text feature passes through a single transformer layer to generate deeper text embeddings $l'_{cls}$.

For the image feature, VL-PUB employs a pyramid feature module with multi-stages to generate image feature with various scales after the fusion with text embeddings. Following Lin et al. (2017); Li et al. (2022b), we leveraged Deconvolution layers and MaxPooling layers for the upscaling and downscaling of feature maps, resulting in feature maps with 5 different scales.

## 3.3 VISUAL-LANGUAGE REGION PROPOSAL NETWORK

Since RPN in OD aims to propose regions for any class, whether base or novel, we emphasize the importance of the RPN module for OVD in this section. Given previous success of region-based visual-language alignment for region classification (Zhong et al., 2022; Li et al., 2022a) and the superority of sigmoid loss over softmax contrastive loss (Zhai et al., 2023), we propose a novel SigRPN block with sigmoid-based visual-language alignment to enhance the model's generalization capability to novel classes.

Specifically, a VL-Fuse layer first fuses the text feature with all visual features of various scales using bi-directional cross-attention to enhance context understanding and generate more informative feature representations as follows,

$$v'_f, l'_f = \text{VLFuse}(\text{Concat}(v_1, v_2, v_3, v_4, v_5), l'_{cls}) \tag{3}$$

where $v'_f$, $l'_f$ denotes the fused visual feature and language feature separately after the VL-Fuse layer of SigRPN. And the text feature will go through a single transformer layer and generate more contextual embeddings $l''_{cls}$.

For the visual feature maps, two convolution layers first enhance the visual representation for each scale. We then fed the enhanced visual features to two parallel convolution layers to generate *object-ness logits* and *anchor deltas* per anchor. *objectness logits* are used for proposal binary classification to determine whether this anchor belongs to background or foreground objects in a contrastive way. *anchor deltas* are used to transform anchors into box proposals.

## 3.4 MULTI-LEVEL FINE-GRAINED ALIGNMENT LOSS

To enable more robust visual-language alignment, we introduce alignment loss into multiple stages and effectively combine them to achieve better performance.

Table 1: VLDet outperforms all baselines for $Novel$ categories for OVD on COCO2017 and LVIS.

| Method | Pre-train Data | Backbone | COCO2017 | | | LVIS | | | |
|---|---|---|---|---|---|---|---|---|---|
| | | | $AP_{Novel}$ | $AP_{Base}$ | $AP_{All}$ | $AP_r$ | $AP_c$ | $AP_f$ | AP |
| OVR-CNN (Zareian et al., 2021) | COCO Caption | RN50 | 22.8 | 46.0 | 39.9 | - | - | - | - |
| ViLD (Gu et al., 2021) | CC3M | RN50 | 27.6 | **59.5** | 51.3 | 16.7 | 26.5 | 34.2 | 27.8 |
| RegionCLIP (Zhong et al., 2022) | CC3M | RN50 | 31.4 | 57.1 | 50.4 | 17.1 | 27.4 | 34.0 | 28.2 |
| F-VLM (Kuo et al., 2022) | CLIP | RN50 | 28.0 | - | 39.6 | 18.6 | - | - | 24.2 |
| BARON (Wu et al., 2023a) | CLIP+COCO Cap | RN50 | 42.7 | 54.9 | 51.0 | 23.2 | 29.3 | 32.5 | 29.5 |
| YOLO-World-M (Cheng et al., 2024) | O365+Gold | Y8-M | - | - | - | 15.9 | 24.6 | 39.0 | 28.8 |
| YOLO-World-L (Cheng et al., 2024) | O365+Gold+CC3M | Y8-L | - | - | - | 20.4 | 31.1 | 43.5 | 34.1 |
| OV-DQUO (Wang et al., 2025) | CLIP | RN50x4 | 45.6 | - | 48.1 | - | - | - | - |
| CCKT-Det++ (Zhang et al., 2025) | CLIP | SwinB | 46.0 | - | 46.2 | - | - | - | - |
| **VLDet-B** (Ours) | O365 | ViT-B | 54.2 | 50.4 | 51.2 | 16.9 | 30.1 | 43.8 | 35.6 |
| **VLDet-L** (Ours) | O365 | ViT-L | **58.7** | 53.9 | **55.2** | **24.8** | **44.7** | **48.0** | **44.6** |

**Mini-Batch Image Contrastive Loss.** To fully exploit the pre-aligned image and text encoders from CLIP, we absorb the conventional CLIP training loss into our VLDet framework for image-wise alignment. Although CLIP recommends a large batch size for contrastive learning, we found it results in worse detection performance since it leads to severe loss balance issues and emphasis too much on the image-wise visual-language alignment compared with RPN and ROI losses. To reduce the scale of image-wise contrastive loss and prevent it from dominating the training process, we divide inputs into mini-batches and compute the image-wise contrastive loss as follows,

$$\mathcal{L}_{ICL} = -\frac{1}{2B}\sum_{k=1}^{B/M}\sum_{m_k=1}^{M}(log\frac{exp(\phi(\boldsymbol{v}_{m_k},\boldsymbol{l}_{m_k})/\tau)}{\sum_{n_k=1}^{M}exp(\phi(\boldsymbol{v}_{m_k},\boldsymbol{l}_{n_k})/\tau)} + log\frac{exp(\phi(\boldsymbol{l}_{m_k},\boldsymbol{v}_{m_k})/\tau)}{\sum_{n_k=1}^{M}exp(\phi(\boldsymbol{l}_{m_k},\boldsymbol{v}_{n_k})/\tau)})$$

(4)

where $B$ is the batch size, and $M$ is a hyper-parameter denoting the mini-batch size. During our experiment, we found $M = 8$ yields the best trade-off between the losses. $\phi(\boldsymbol{v},\boldsymbol{l})$ is the cosine similarity between visual and language embeddings and $\tau$ is the temperature (Wu et al., 2018).

**Anchor-Text Binary Alignment Loss.** Besides the image-wise contrastive loss that aligns the image feature and text feature in a coarse-level, we introduce two region-wise alignment losses for fine-grained alignment. The first one is Anchor-text binary Alignment Loss (AAL) within the SigRPN module. Conventional RPNs typically solve the background-vs-foreground problem via a binary classification. In SigRPN, we tackle this problem in a constrastive manner. Specifically, we first compute the similarities of visual features with text features of all classes, including *background*. We then use the mean of similarities over all object classes to subtract the similarity with *background* as the final *objectness score* $\boldsymbol{s}_{obj}$ as follows,

$$\boldsymbol{s}_{obj} = (\frac{1}{N'}\sum_{i=1}^{N'}\phi(\boldsymbol{v},\boldsymbol{l}_i) - \phi(\boldsymbol{v},\boldsymbol{l}_0))/\tau$$

(5)

where $N'$ denotes the number of foreground object categories, and $\boldsymbol{l}_0$ is the text feature of *background*. We wrap $\boldsymbol{s}_{obj}$ within a binary cross entropy (BCE) loss as the RPN classification loss $\mathcal{L}_{AAL}$.

**Region-Text Alignment Loss.** For ROI, we wrap the similarity of visual features and text features into categorical cross entropy as region-text alignment loss $\mathcal{L}_{RAL}$ by following Li et al. (2022a).

## 4 EXPERIMENTAL RESULTS

**Datasets.** Following previous works (Zareian et al., 2021; Gu et al., 2021; Zhong et al., 2022; Cheng et al., 2024), we evaluated our approach on two public datasets: COCO2017 (Lin et al., 2014) and LVIS (Gupta et al., 2019). To evaluate the performance of open-vocabulary object detection, we follow the standard protocol to split the object categories into *Base* and *Novel* sets, train the model with only annotations of the *Base* categories, and evaluate on *Base*, *Novel*, and *All* all together. For COCO2017, We follow the data split of Zareian et al. (2021) with 48 *Base* categories and 17 *Novel* categories, and *ALL* set contains 65 categories., which yields 107,761 training images and 4,836 test images. On LVIS, following Gu et al. (2021), we adopt the category split with 866 *Base* categories (common and frequent objects) and 337 *Novel* categories (rare objects). Additionally, VLDet is

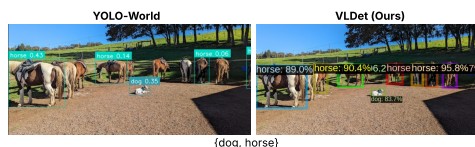 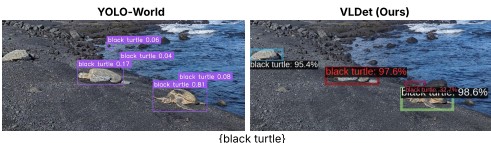

Figure 4: Comparison between YOLO-World and VLDet, VLDet detects objects (*e.g.*"black turtle") more accurately. More visualized results and illustrations are in Sec. B.

pretrained with only one public dataset, Objects365, with captions generated by BLIP-2 through prompt, "Describe this image in one sentence.", containing about 1.7M image-caption pairs.

**Implementation Details.** We adopted the two-stage Cascade RCNN (Cai & Vasconcelos, 2019) as our basic detection framework. We employed ViT-B and ViT-L image encoders and the corresponding text encoders from CLIP as our image encoder and text encoder, respectively. During pre-training, we adopted the learning rate of $1 \times 10^{-5}$ for language backbone and $1 \times 10^{-4}$ for the rest modules. We configured the AdamW optimizer with weight decay of $1 \times 10^{-4}$. Batch size is 2 for our VLDet-B and 1 for VLDet-L. In the OVD fine-tuning stage, we reduced the learning rate to $1 \times 10^{-5}$ for all layers and froze the visual-to-language projection layers (V2L), while leaving other settings unchanged. The model is trained for a maximum of 50 epochs during pre-training and 15 epochs during fine-tuning. In addition, we set the maximum token length for captions to 64 and pad class names to the longest one (*e.g.*5 tokens for each class on COCO2017). All experiments are conducted on 64 NVIDIA A100 GPUs.

### 4.1 VLDET ON OPEN-VOCABULARY OBJECT DETECTION

We evaluate VLDet on COCO2017 and LVIS for OVD after pre-training on Objects365, the detailed comparison with baselines are shown in Tab. 1. When experimenting on COCO2017 and LVIS, we train with only the *Base* categories (*Common* and *Frequent* for LVIS) and evaluate on different sets of categories.

Compared to all baselines, VLDet-L, utilizing the ViT-L backbone, achieves the highest AP across *All* categories. More importantly, it also attains the best performance on *Novel* categories, elevating the AP from 46.0 of CCKT-Det++ (Zhang et al., 2025) to 58.7 on COCO2017 and from 23.2 of BARON (Wu et al., 2023a) to 24.8 on LVIS. This demonstrates its extraordinary capability for generalization on new categories. Notably, for COCO2017, which contains 17 *Novel* classes, the AP of our VLDet on *Novel* categories surpasses that of *Base* categories. Additionally, VLDet-B also achieves superior performance, further highlighting the benefits of our multi-level alignment loss for visual-language alignment. ViLD (Gu et al., 2021) achieves the best AP on *Base* classes with data augmentation through large-scale jittering (Ghiasi et al., 2021) and a significantly longer training schedule. Given that LVIS contains 337 *Novel* categories, far more than COCO2017, VLDet-B with the ViT-B backbone performs limitedly. However, our larger model, VLDet-L outperforms all baselines across all metrics. Notably, when compared to YOLO-World-L (Cheng et al., 2024), which is pre-trained on a larger scale of data, our VLDet enhances the AP for *Novel* categories by 4.43, demonstrating the superiority of the two-stage framework and SigRPN for OVD. We show the visualized comparison in Fig. 4.

### 4.2 VLDET ON CLOSED-SET OBJECT DETECTION

For closed-set OD, we evaluated two scenarios: first, we conducted a zero-shot evaluation of our pre-trained model on the COCO2017 dataset (ZOD); second, we assessed the performance of our model after fine-tuning (FOD).

As shown in Tab. 2, for ZOD, VLDet-L achieves superior performance with an AP of 45.8, surpassing baselines pre-trained on data of similar or even larger scale, such as YOLO-World (Cheng et al., 2024), which includes pseudo-labeled bounding boxes on CC3M and utilizes a vast category set extracted from grounding dataset captions. This demonstrates the efficacy of our framework, which adapts pre-aligned image and text encoders from CLIP with enriched visual-language semantic space and integrates a more robust visual-language alignment strategy. VLDet is more efficient, requiring 221 GLOPs for introduced layers versus 407 GLOPs for VLDyHead in GLIP (Li et al.,

Table 2: Results for Zero-Shot Object Detection (ZOD) and Fine-Tuned Object Detection (FOD) on the COCO2017 dataset demonstrate that our VLDet model exhibits strong capabilities in both cases with pre-training solely on the Objects365 dataset. Numbers in gray mean that they are not in zero-shot manner with pre-training data including COCO.

| Method | Pre-train Data | ZOD (AP) | FOD (AP) |
|---|---|---|---|
| ViTDet-B (Li et al., 2022b) | IN1K | - | 51.6 |
| YOLOv11-L (Jocher & Qiu, 2024) | - | - | 53.4 |
| ViLD (Gu et al., 2021) | CLIP400M | 36.6 | 39.1 |
| RegionCLIP (Zhong et al., 2022) | CC3M | 29.6 | - |
| GLIP-T (Li et al., 2022a) | O365+Gold | 44.6 | 53.8 |
| YOLO-World-M (Cheng et al., 2024) | O365+Gold | 42.8 | 51.2 |
| YOLO-World-L (Cheng et al., 2024) | O365+Gold+CC | 45.1 | 53.3 |
| FIBER-B (Dou et al., 2022) | O365+COCO+CC+SBU+VG | 49.3 | 58.4 |
| Grounding-DINO-L (Liu et al., 2023) | O365+COCO+OI+Gold+Cap4M+RefC | 60.7 | 62.6 |
| **VLDet-B** (Ours) | O365 | 43.7 | 54.1 |
| **VLDet-L** (Ours) | O365 | **45.8** | **55.9** |

Table 3: $\mathcal{L}_{ICL}$ and $\mathcal{L}_{AAL}$ contribute to enhanced detection performance, with $\mathcal{L}_{AAL}$ providing greater improvements for rare and common categories.

| $\mathcal{L}_{RAL}$ | $\mathcal{L}_{ICL}$ | $\mathcal{L}_{AAL}$ | $AP_r$ | $AP_c$ | $AP_f$ | AP |
|---|---|---|---|---|---|---|
| ✓ | ✗ | ✗ | 12.43 | 22.65 | 42.93 | 31.56 |
| ✓ | ✓ | ✗ | 14.90 | 26.18 | **43.87** | 33.76 |
| ✓ | ✓ | ✓ | **16.93** | **30.14** | 43.84 | **35.62** |

2022a). Furthermore, for FOD, VLDet model also exhibits superior performance, achieving an AP of 55.9 after fine-tuning within 5 epochs. This surpasses not only open-world object detectors but also the state-of-the-art traditional object detector, YOLOv11 (Jocher & Qiu, 2024). We also include results from FIBER-B (Dou et al., 2022) and Grounding-DINO-L (Liu et al., 2023), which report great zero-shot performance. However, they are pre-trained on a dataset of significantly larger scale, even including COCO (thereby not strictly qualifying as zero-shot inference).

### 4.3 Ablation Studies

**Multi-Level Alignment Loss.** In Tab. 3, we show the fine-tuning results of VLDet-B on LVIS dataset for open-vocabulary object detection (OVD) with different losses. We can find that both of our introduced image-level loss $\mathcal{L}_{ICL}$ and anchor-text binary alignment loss $\mathcal{L}_{AAL}$ can enhance not only the overall AP on all classes, but also the AP on *Novel* categories. These results demonstrate that the multi-level loss is beneficial for visual-language alignment. Specifically, the mini-batch image contrastive loss elevates the AP for $Novel$ categories of the LVIS dataset from 12.43 to 14.90 by 2.47, and then SigRPN further improves it to 16.93 by 2.03. In addition, the overall AP in all categories also improved from 31.56 to 35.62.

Importantly, with SigRPN and $\mathcal{L}_{AAL}$, we observe that the AP for frequent classes remains consistent, while the AP for rare and common classes significantly increases. Notably, our training data does not include the rare categories, highlighting the exceptional capability of our SigRPN in generalizing to *Novel* categories by enabling the RPN to binary-differentiate background and object of any class with visual-language multi-modal embedding space.

We compare the performance of VLDet-B on standard object detection tasks using the COCO2017 dataset, trained from scratch with varying mini-batch sizes to balance image-wise visual-language alignment with final detection accuracy. We find that a mini-batch size of 8 yields the best performance, achieving an AP of 40.98. Detailed results are in Sec. C.

**Specific Design of Our VL-Encoder.** In this section, we detail the design of our VL-Encoder by experimenting with our VLDet-B for traditional object detection on the COCO2017 dataset from

Table 4: Results of different variants demonstrate the efficacy of our design.

| Model | AP | $AP_{50}$ | $AP_{75}$ |
|---|---|---|---|
| Not pre-aligned Backbones | 34.86 | 55.96 | 36.78 |
| Pre-aligned, Frozen $E_L$ | 37.29 | 58.46 | 39.85 |
| Pre-aligned, Active $E_L$ | **38.03** | **59.29** | **40.91** |
| Single-scale Backbone | 35.78 | 53.13 | 38.6 |
| Multi-scale Backbone | **38.03** | **59.29** | **40.91** |
| Plain model (Only class prompt) | 38.03 | 59.29 | 40.91 |
| + Caption | 38.89 | 59.47 | 41.76 |
| + Caption & $\mathcal{L}_{ICL}$ | 40.18 | 62.07 | 43.22 |
| + Caption & $\mathcal{L}_{ICL}$ & VL-Fuse | **40.98** | **62.58** | **44.25** |
| Class-template prompt (VLDet) | 40.88 | 62.68 | 44.33 |
| Class-name prompt (VLDet) | 40.98 | 62.58 | 44.25 |

Table 5: Freeze different layers for fine-tuning on OVD. $E_L$ denote the language encoder, $V2L_1$ denote the final logit projection layer in SigRPN, and $V2L_2$ denote the logit projection layer in ROI.

| $E_L$ | $V2L_1$ | $V2L_2$ | AP | $AP_r$ | $AP_c$ | $AP_f$ |
|---|---|---|---|---|---|---|
| ✗ | ✗ | ✗ | 44.03 | 14.77 | **45.44** | **48.04** |
| ✗ | ✗ | ✓ | 44.09 | 24.24 | 43.85 | 47.88 |
| ✗ | ✓ | ✓ | **44.33** | **24.83** | 44.33 | 47.84 |
| ✓ | ✗ | ✓ | 40.82 | 24.65 | 37.17 | 46.97 |
| ✓ | ✓ | ✓ | 41.78 | 24.26 | 38.89 | 47.49 |

scratch. Initially, we replaced the pre-aligned text encoder for ViT-B/16 from CLIP with ViT-B/32, while retaining ViT-B/16 as the image encoder. We observed that the pre-aligned backbones, even when frozen, provide significantly higher AP, highlighting the superiority of inheriting enriched pre-aligned semantic information from CLIP. Subsequently, we found that employing a multi-scale backbone with VL-PUB, as opposed to a single-scale model, improves the AP from 35.78 to 38.03. This demonstrates that pyramid features enhance detection accuracy by providing more spatial information. Furthermore, by comparing our model trained with and without captions, we discovered that incorporating captions alongside class prompts provides additional contextual information, enhancing the AP from 38.03 to 38.89. Besides, we found that adding $\mathcal{L}_{ICL}$ improves AP from 38.89 to 40.18 and the VL-Fuse layer further increases it to 40.98. Finally, we experimented with the text prompt template from previous work (Radford et al., 2021), such as "A photo of {label}", but observed no performance gain. This approach also requires more tokens compared to using plain class names (*i.e.*only 5 tokens on COCO), introducing additional computational cost. Therefore, we design our class prompts with plain class names.

**Fine-tuning Strategies.** For fine-tuning the pre-trained model on COCO2017 or LVIS for OVD, it is essential to freeze the visual projection layer (*i.e.*V2L) (Zareian et al., 2021). Without this step, the model is prone to overfitting to *Base* classes, resulting in a significant decline in performance on *Novel* categories. As shown in Tab. 5, when no layers are frozen, the AP on *Base* categories (*i.e.*$AP_c$ and $AP_f$) remains high, but $AP_r$ is notably low due to overfitting. We also observed that freezing the language background can lead to some loss in accuracy. The optimal fine-tuning strategy involves freezing the two V2L layers.

## 5 CONCLUSION

We successfully extend the pre-aligned image and text encoders from CLIP, with its extensive visual-language latent space, into a novel OVD framework, Visual-Language Detection (VLDet). VLDet preserves the structure of feature pyramid, following the fusion of visual and language features, to boost spatial signals. We further introduce three constrastive losses to achieve the multi-level, fine-grained visual-language alignment, resulting superior detection performance on the OVD scenario. VLDet significantly elevates the AP on novel categories by 27.6% on COCO2017 and 6.9% on LVIS, and outperforms zero-shot object detection baselines when pre-trained solely on Objects365.

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

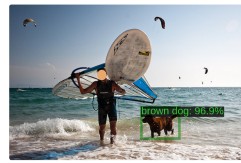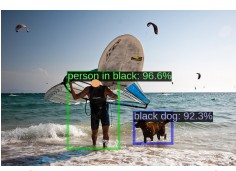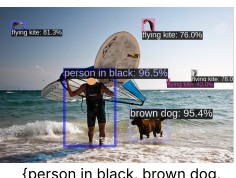

{dog}          {brown dog}          {person in black, black dog}          {person in black, brown dog, flying kite}

Figure 5: **Visualized Results for Referring Object Detection.** We demonstrate VLDet's ability to generalize to novel categories by providing descriptive noun phrases (e.g., "brown dog") as input instead of just class names.

# APPENDIX

## A  BI-DIRECTIONAL CROSS-ATTENTION

Bi-directional Cross-attention is widely used for vision-language feature fusion (Li et al., 2022a; Liu et al., 2023). Specifically, the mechanism treats one modality as the Query and the other as the Key/Value. This occurs in two directions:

- **Visual-to-Language**: The visual features attend to the language features to incorporate semantic context.

$$\boldsymbol{v}_{out} = \boldsymbol{v} + \text{MHCA}(Q = \boldsymbol{v}W_q^{\boldsymbol{v}}, K = \boldsymbol{l}W_k^{\boldsymbol{l}}, V = \boldsymbol{l}W_v^{\boldsymbol{l}}) \tag{6}$$

- **Language-to-Visual**: The language features attend to the visual features to ground the text in the specific image content.

$$\boldsymbol{l}_{out} = \boldsymbol{l} + \text{MHCA}(Q = \boldsymbol{l}W_q^{\boldsymbol{l}}, K = \boldsymbol{v}W_k^{\boldsymbol{v}}, V = \boldsymbol{v}W_v^{\boldsymbol{v}}) \tag{7}$$

where $W$ terms are learnable projection matrices and MHCA denotes multi-head cross-attention. This mechanism is applied in Eq. (2) for single-scale features and Eq. (3) for multi-scale features.

## B  ADDITIONAL VISUALIZATION RESULTS

**Visualized Results for Referring Object Detection.** To demonstrate the generalization capability of VLDet to novel object categories, we perform inference on the same image multiple times using various category names. We deliberately include descriptive words to form noun phrases (*e.g.*"brown dog"). As shown in Fig. 5, VLDet consistently and successfully detects the target objects, even with these modified novel category names, and achieves high confidence scores.

**Visualized Comparison to YOLO-World.** In Fig. 6, we provide two additional, higher-resolution comparison examples against YOLO-World, supplementing the results in the main text. We again use customized user vocabularies to evaluate generalization. For the first sample, the vocabulary is defined as "person in black, brown dog, flying kite", and for the second, it is "person in jacket, refrigerator, cup". The visualized results clearly demonstrate that VLDet detects all objects more accurately. In contrast, YOLO-World failed to detect the "person in black" in the first sample (after the descriptive "in black" was added) and detected only one "person in jacket" in the second sample, whereas VLDet successfully detected both.

**More Visualized Results in Diverse Scenarios.** Finally, to further showcase the capabilities of VLDet, we present results from multiple diverse scenarios in Fig. 7. These visualizations further demonstrate the superiority of VLDet in accurately detecting a wide variety of objects. The examples show strong performance across different object categories (including fruits, plants, animals, vehicles, person *etc*.) and at various object scales.

**YOLO-World**  **VLDet (Ours)**

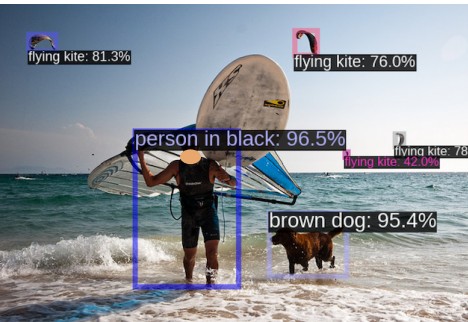

{person in black, brown dog, flying kite}

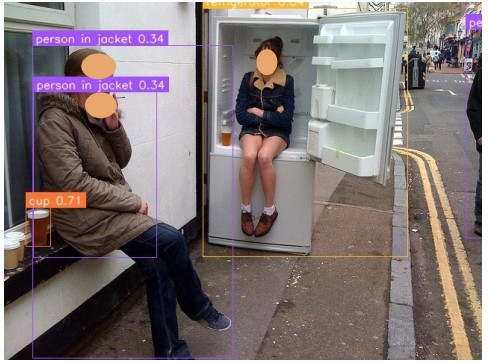
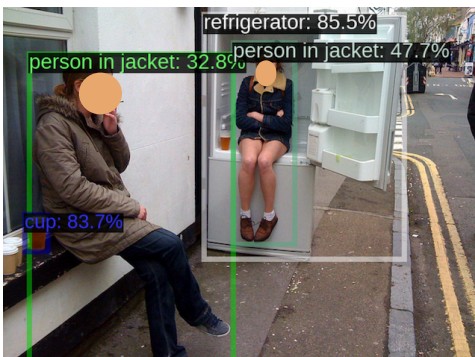

{person in jacket, refrigerator, cup}

Figure 6: **Additional Visual Comparisons to YOLO-World.** VLDet provides more accurate detection results on custom user vocabularies (e.g., "person in black," "person in jacket") compared to the baseline.

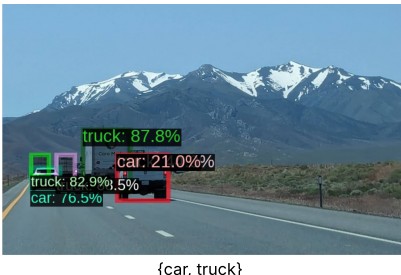
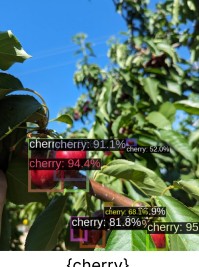
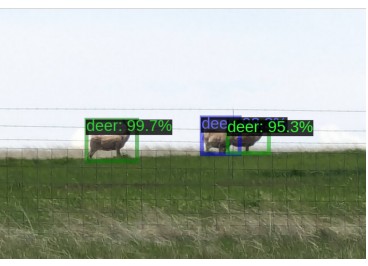

{car, truck}  {cherry}  {deer}

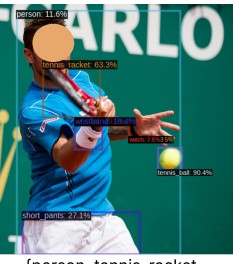
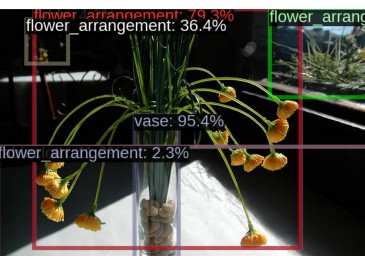
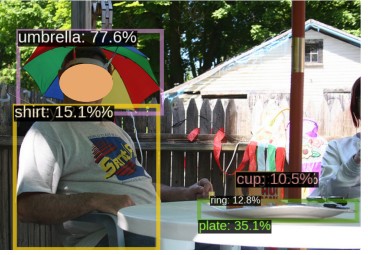

{person, tennis_racket, tennis_ball, watch, wristband, short_pants}  {flower_arragement, vase}  {umbrella, plate, cup, shirt, ring}

Figure 7: **More Visualization Results on Custom Vocabularies.** VLDet successfully detects objects across diverse scenarios, categories, and scales.

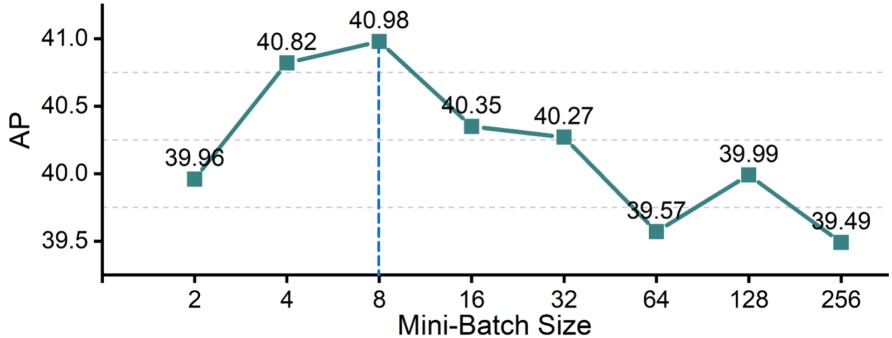

Figure 8: Image-level Contrastive Loss, $\mathcal{L}_{ICL}$ performs the best with mini-batch size of 8.

Table 6: VLDet outperforms self-distilled baseline for $Novel$ categories with various model sizes.

| Method | Pre-train Data | ViT-B | | | ViT-L | | |
| --- | --- | --- | --- | --- | --- | --- | --- |
| | | $AP_{Novel}$ | $AP_{Base}$ | $AP_{All}$ | $AP_{Novel}$ | $AP_{Base}$ | $AP_{All}$ |
| F-ViT+CLIPSelf (Wu et al., 2023b) | CLIP | 37.6 | - | - | 44.3 | - | - |
| **VLDet** (Ours) | O365 | 54.2 | 50.4 | 51.2 | 58.7 | 53.9 | 55.2 |

## C   EXPLORATION ON MINI-BATCH SIZE

While previous work, such as CLIP, suggests that large batch sizes work better for contrastive learning, they may dominate the training process and degrade the final detection performance.

To balance image-wise and region-wise visual-language alignment, we explore a range of mini-batch sizes. Our experiments demonstrate that a batch size of 8 provides the best performance, as shown in Fig. 8.

## D   COMPARED TO SELF-DISTILLED BASELINE

In the main text, our comparisons primarily focused on baselines that use ResNet as the vision backbone and rely on knowledge distillation to transfer visual-language (VL) knowledge from large VLMs like CLIP.

Here, we provide an additional, critical comparison with a state-of-the-art method that also directly leverages the Vision Transformer (ViT) from pre-aligned VLMs: CLIPSelf (Wu et al., 2023b). CLIPSelf employs a self-distillation technique to refine the dense feature map of a CLIP ViT, making it more suitable for localization in dense prediction tasks. This approach enables the direct use of ViT backbones for open-vocabulary object detection (OVD).

As shown in Tab. 6, VLDet significantly outperforms CLIPSelf on COCO novel categories benchmarks with both ViT-B and ViT-L as vision backbone. This consistently superior performance further validates the effectiveness and robustness of our framework by adapting pre-aligned vision encoder into OVD detector through multi-level fine-grained contrastive loss.

