# OpenReview forum: "Enhancing Open-Vocabulary Object Detection through Multi-Level Fine-Grained Visual-Language Alignment"
_ICLR.cc/2026/Conference — Submitted to ICLR 2026_

### Official Review · Reviewer_zJjL · 2025-10-30

**Soundness:** 4
**Presentation:** 2
**Contribution:** 3
**Rating:** 4
**Confidence:** 3

**Summary:**

This paper identifies that previous open-vocabulary object detection methods face challenges in adapting single-scale image backbones for object detection tasks. It further introduces a fine-grained visual-language alignment framework to bridge the image-region gap.

**Strengths:**

**Originality**:
- This paper's motivation is sound, as current open-vocabulary object detection methods have not adequately addressed the challenge of multi-scale features.

**Clarity**:
- This paper clearly explains its motivation and methodology.

**Significance**:
- This paper addresses the problem of region-text misalignment, proposes a reasonable method, and proves its effectiveness through experiments.

**Weaknesses:**

- The definition of "fine-grained" in this paper is unclear. If it means region-level alignment (fine-grained) compared with image-level alignment (coarse-grained), I don't think that region-level alignment is enough to be called 'fine-grained alignment'.
- In Table 1, the backbones of compared methods are different from the backbone used in this paper, which leads to an unfair comparison. Some methods that also use ViT as the backbone should be included.
- The visualization examples in this paper are very rough and can not clearly show the effect they are intended to demonstrate.

**Questions:**

- I'm wondering about what "fine-grained" exactly means in this paper, or how it is different than what is called "coarse-grained"?

---

> ### Author Response · Authors · 2025-11-24
>
> We thank the reviewer for their valuable feedback and for recognizing the excellent soundness, clear methodology, and significant problem contribution of our work.
> We are encouraged that the reviewer finds our motivation sound and our method reasonable. Below, we address each of the concerns and questions raised:
>
> > The definition of "fine-grained" in this paper is unclear. If it means region-level alignment (fine-grained) compared with image-level alignment (coarse-grained), I don't think that region-level alignment is enough to be called 'fine-grained alignment'.
>
> > what "fine-grained" exactly means in this paper, or how it is different than what is called "coarse-grained"?
>
> We apologize for the ambiguity. In the vision–language literature, coarse-grained alignment typically refers to image-level matching with full sentences (e.g., captions), whereas "fine-grained" is often used for region-level alignment with words (e.g., object categories) [1,2,3].
> In our context, "fine-grained" refers to the hierarchy of alignment:
> 1) Image-Level Alignment (Coarse). We use a Mini-Batch Image Contrastive Loss ($L_{ICL}$) to preserve the coarse, image-to-caption alignment from the original VLM.
> 2) Anchor-Level Alignment (Fine). This is our core novelty. Our SigRPN module introduces an Anchor-Text Binary Alignment Loss ($L_{AAL}$). This loss operates at the anchor level within the RPN, teaching it to distinguish "background" from "any object" using visual-language alignment before any regions are even proposed.
> 3) Region-Level Alignment (Fine). We use a standard Region-Text Alignment Loss ($L_{RAL}$) in the ROI Head, which aligns the features of proposed regions to class text.
>
> Therefore, our use of “fine-grained” does not merely refer to region-level alignment; it refers to this three-level alignment hierarchy, with the new anchor-level alignment providing additional semantic supervision than prior work. This is what provides the robust generalization shown in our ablation studies (Table 3).
>
>
> > In Table 1, the backbones of compared methods are different from the backbone used in this paper, which leads to an unfair comparison. Some methods that also use ViT as the backbone should be included.
>
> 1)	We included these foundational methods (like ViLD, RegionCLIP, BARON) because they are the canonical, most-cited SOTA baselines in OVD. Omitting them would make it difficult to contextualize our work and demonstrate progression in the field
> 2)	There is a specific reason why many foundational OVD methods, such as ViLD, RegionCLIP, and BARON, rely on ResNet backbones. This was not an arbitrary choice by those authors. As we discuss in our paper, adapting pre-aligned VLMs like CLIP for detection has been a major challenge because their ViT image encoders are single-scale , whereas detection fundamentally requires multi-scale feature pyramids to handle objects of various sizes. Previous works, like ViLD, had to work around this by distilling knowledge from a ViT into a ResNet backbone. As one key contribution of our paper, we propose a novel, unified framework to adapt a pre-aligned, single-scale ViT directly into a multi-scale OVD architecture. Here, we show the superiority of our direct adaptation method over the previous, more complex distillation-based workarounds:
>  | Method | COCO2017 (AP_Novel) | LVIS (AP_Novel) |
> | :------- | :------: | -------: |
> | ViLD (distill from ViT)     | 27.6| 16.7|
> | BARON (distill from ViT)   | 42.7  | 23.2  |
> | VLDet-B (ours)   | 58.72   | 24.82   |
>
>
> > The visualization examples in this paper are very rough and can not clearly show the effect they are intended to demonstrate.
>
> We appreciate the feedback on our qualitative examples. The intent of Figure 4 was to provide a direct comparison to the very strong YOLO-World baseline. For example, our VLDet successfully and more accurately identifies multiple "black turtle", a novel and challenging category, compared to YOLO-World, and our model provides a much higher confidence score (98.6%) than YOLO-World (81.0%), demonstrating the enhanced generalization our method provides.
> We take the reviewer's point on clarity and will add more comprehensive and visually polished qualitative examples to the appendix in the revised version.
>
> [1] Zhong, Yiwu, et al. "Regionclip: Region-based language-image pretraining." Proceedings of the IEEE/CVF conference on computer vision and pattern recognition. 2022.
>
> [2] Li, Liunian Harold, et al. "Grounded language-image pre-training." Proceedings of the IEEE/CVF conference on computer vision and pattern recognition. 2022.
>
> [3] Liu, Shilong, et al. "Grounding dino: Marrying dino with grounded pre-training for open-set object detection." European conference on computer vision. Cham: Springer Nature Switzerland, 2024.

---

> > ### Comment · Reviewer_zJjL · 2025-11-25
> >
> > The authors have addressed most of my concerns.
> >
> > I hope the author can include these information into the main text, especially the explanation of "fine-grained alignment".
> >
> > In addition, regarding the issue of visualizations, the author only replied in text. I hope to see higher-quality images in the revised version, including Figure 4 and the appendix.

---

> > > ### Author Response · Authors · 2025-11-26
> > >
> > > We are delighted to hear that we have addressed most of your concerns. Thank you for your valuable suggestions for improving the clarity and presentation of our work.
> > >
> > > We have incorporated your feedback into the revised paper as follows:
> > > 1) Explanation of "fine-grained alignment": We have polished the main text and included a detailed explanation of "fine-grained alignment" (**lines 68-77**) to clarify its definition and significance in our framework.
> > > 2) Higher-Quality Visualizations: We have regenerated **Figure 4** in the main text with higher resolution and clarity. Furthermore, we have added three new groups of detailed visualization results to the **Appendix B** (All figures are provided in **.pdf** format; please feel free to zoom in for clear details):
> > > - Referring Object Detection (Figure 5): Demonstrates VLDet's strong generalization to novel categories by using different descriptive noun phrases (e.g., "brown dog") for the same input image.
> > > - Comparison to YOLO-World (Figure 6): Augmenting the two examples in the main text (Figure 4), we provide more high-resolution comparisons that consistently highlight VLDet's superiority in object localization and classification accuracy.
> > > - Diverse Scenarios (Figure 7): We include a broader set of qualitative results showcasing VLDet's robust performance across various object categories and scales in challenging, real-world scenes.
> > >
> > > We believe these additions significantly enhance the clarity and impact of our paper. Thank you again for your constructive review. If you have any more comments, please let us know.

---

### Official Review · Reviewer_JqGJ · 2025-10-31

**Soundness:** 3
**Presentation:** 4
**Contribution:** 3
**Rating:** 6
**Confidence:** 3

**Summary:**

This paper presents VLDet, a framework for Open-Vocabulary Object Detection (OVD) that enhances fine-grained visual-language alignment by effectively adapting CLIP encoders for detection tasks.

Previous OVD methods either use CLIP’s single-scale image encoder, which limits spatial precision, or train multi-scale backbones from scratch using pseudo-labeled data from zero-shot detectors such as GLIP, resulting in inefficiency and bias. Moreover, they emphasize region-level alignment while neglecting image-level alignment, which is also crucial for robust visual-language modeling.

The main contributions are as follows:

- Visual-Language Pyramid Upscale Block (VL-PUB) adapts CLIP’s image and text encoders into an open-world detector, leveraging multi-scale image features for improved detection accuracy.

- Sigmoid-based Region Proposal Network (SigRPN) introduces sigmoid-based visual-language contrastive learning to differentiate background from objects of any class and integrates image-wise contrastive loss for stronger alignment.

- Extensive Experiments show that VLDet achieves state-of-the-art performance on COCO2017 and LVIS, improving novel class mAP from 42.7 to 58.72 and from 23.2 to 24.83, demonstrating strong generalization to unseen categories.

Overall, the paper proposes a unified and effective approach for adapting CLIP to open-vocabulary detection by improving spatial precision, cross-modal alignment, and generalization.

**Strengths:**

- The paper is clearly written and logically structured, with a well-motivated problem and methodology.

- The proposed VL-PUB and SigRPN modules directly address critical limitations of prior OVD methods.

- Extensive experiments on standard benchmarks show consistent and significant performance improvements.

**Weaknesses:**

- The introduction could be better organized to separate the problem definition, motivation, and main contributions.

- The VL-Fuse layer employs bi-directional cross-attention, but its mechanism is not clearly explained or supported with mathematical details.

- The paper lacks comparisons with more recent OVD baselines that use stronger backbones or alignment mechanisms.

**Questions:**

- In the VL-Fuse Layer, why do the text features differ between the first and second fusion stages (from $l_{cls}$ and $l_{cap}$ to $l'_{cls}$), and what is the intuition behind this design choice?

- What is the rationale for subtracting the mean similarity across all object classes when computing background similarity, and how does this normalization improve detection stability?

- Can VLDet be applied to other open-vocabulary detectors or backbones beyond CLIP’s ViT architecture?

- How does the proposed sigmoid-based loss differ from previous sigmoid or softmax approaches, and why does it achieve better alignment for novel object detection?

---

> ### Author Response · Authors · 2025-11-24
>
> Thank you for your constructive feedback and for recognizing that our paper is clearly written, well-motivated, and achieves strong experimental results. We appreciate that you see the value of VL-PUB and SigRPN in addressing critical limitations of prior methods. We would like to answer your excellent questions:
>
> > The introduction could be better organized to separate the problem definition, motivation, and main contributions.
>
> Thank you for the suggestion. We will revise the introduction in the revised version to more clearly separate the problem definition, our motivation, and the list of contributions.
>
> > The VL-Fuse layer employs bi-directional cross-attention, but its mechanism is not clearly explained or supported with mathematical details.
>
> Specifically, the mechanism treats one modality as the Query and the other as the Key/Value. This occurs in two directions:
> - Visual-to-Language: The visual features attend to the language features to incorporate semantic context.$$V_{out} = V + \text{MHCA}(Q=V W_q^v, K=L W_k^l, V=L W_v^l)$$
> - Language-to-Visual: The language features attend to the visual features to ground the text in the specific image content.$$L_{out} = L + \text{MHCA}(Q=L W_q^l, K=V W_k^v, V=V W_v^v)$$where $W$ terms are learnable projection matrices. This mechanism is applied in Eq. (2) for single-scale features and Eq. (3) for multi-scale features. We will include these specific formulations in the Appendix of the revised version.
>
> > The paper lacks comparisons with more recent OVD baselines that use stronger backbones or alignment mechanisms.
>
> We compared against the most recent strong OVD baselines: YOLO-World (CVPR 2024) and OV-DQUO (AAAI 2025), which utilize a larger dataset mixture (e.g., GoldG and CC3M). To further address the backbone strength concern, our VLDet-B (ViT-B backbone) even outperforms the very recent, highly competitive CCKT-Det (**ICLR 2025**) (Swin-B backbone) on $AP_{Novel}$:
> | Method | Backbone |COCO2017 (AP_Novel) |
> | :------- | :------: | :------: |
> | CCKT-Det (ICLR 2025) [1] | Swin-B |46  |
> | VLDet-B| ViT-B    | 54.17|
>
> > In the VL-Fuse Layer, why do the text features differ between the first and second fusion stages (from $l_{cls}$ and $l_{cap}$ to $l'_{cls}$), and what is the intuition behind this design choice?
>
> The difference reflects a design of progressive visual-language alignment.
> 1.  Stage 1 (VL-PUB): We explicitly introduce caption tokens ($l_{cap}$) alongside class tokens ($l_{cls}$) to fuse with the initial image feature ($v_0$). The intuition is that captions provide richer textual context than class names alone, enhancing the semantic embedding space during training.
> 2.  Stage 2 (SigRPN): The resulting refined text features ($l_{cls}'$) are then fused with the generated pyramid image features. This ensures that by the time the text features reach the proposal network, they have already been "visually conditioned" and enriched by the global image context, leading to more robust region-level alignment.
>
> > What is the rationale for subtracting the mean similarity across all object classes when computing background similarity, and how does this normalization improve detection stability?
>
> The formula $s_{obj} = (\text{mean}(obj_{sim}) - bg_{sim}) / \tau$ is designed to create a robust, open-vocabulary "objectness" score
> Instead of just learning "foreground" (which is an unbounded, open set in OVD), we compute a similarity score against all known object classes and take the mean. This average score represents a "general object" concept
> We then explicitly contrast this "general object" score against the "background" similarity score
> This formulation (Eq. 5) forces the model to learn a feature space where any object is more distinct from the background than the average object. This relative margin provides a stable, normalized signal that generalizes well to novel objects, which should also be closer to the "general object" concept than to "background."
>
> > Can VLDet be applied to other open-vocabulary detectors or backbones beyond CLIP’s ViT architecture?
>
> Yes, VLDet is a general framework. The VL-PUB and SigRPN modules can be applied to any single-scale Vision Transformer that is aligned with a text encoder (e.g., SigLIP).
>
> > How does the proposed sigmoid-based loss differ from previous sigmoid or softmax approaches, and why does it achieve better alignment for novel object detection?
>
> Standard RPNs use binary classification (foreground/background). SigRPN reformulates this as a semantic contrastive task. It computes the similarity between visual anchors and text embeddings (including a "background" text embedding). This forces the RPN to learn what makes an object distinct from the background semantically, rather than just visually, which significantly aids in generalizing to novel categories.
>
> [1] Zhang, Chuhan, et al. "Cyclic Contrastive Knowledge Transfer for Open-Vocabulary Object Detection." arXiv preprint arXiv:2503.11005 (2025).

---

### Official Review · Reviewer_J8mg · 2025-10-31

**Soundness:** 2
**Presentation:** 2
**Contribution:** 2
**Rating:** 2
**Confidence:** 4

**Summary:**

The paper proposed three improvements to tackle the open-vocabulary object detection task. First, the Visual-Language Pyramid Upscale Block extracts multi-level feature pyramid from the single-scale image feature to detect objects at various scales. Second, the Visual-Language Region Proposal Network fuses image features with text embeddings and uses an Anchor-Text Binary Alignment Loss for classification. Third, a Mini-Batch Image Contrastive Loss is used to transfer image-level knowledge. The proposed method is evaluated on  COCO and LVIS.

**Strengths:**

1. The paper provides some visualization examples (Figure 1 and Figure 4) for comparison, showing the differences between various models intuitively.
2. The paper provides detailed ablation studies in Table 3, Table 4, and Table 5, showing that each component of the proposed method is useful and the improvements are orthogonal.
3. The formulas in the paper are well-written, making the proposed method easy to understand.

**Weaknesses:**

1. The paper proposed three independent innovations (Visual-Language Pyramid Upscale Block, Visual-Language Region Proposal Network, and Mini-Batch Image Contrastive Loss) without a unified motivation, making the paper A + B + C.
2. Using a multi-scale feature pyramid is a common practice in object detection. I don't think it is a common problem that needs to be tackled. As ViT-based CLIP produces single-scale features, we can use ViTDet [1,2] to produce a feature pyramid.
3. Vision-language deep fusion is also a common practice in open-vocabulary object detection. Authors should compare the proposed VLFuse with related works [3,4,5,6]. I don't think the challenges mentioned from Line 164 to Line 167 are a big problem. All these methods work well with high performance.
4. The experiments in Table 1 are extremely unfair as the proposed method uses much more pretraining data (objects365) and a significantly larger backbone (ViT-Large). Authors should include other strong baselines for comparison, including but not limited to [2, 3]. Further, objects365 contains all 80 classes in COCO. Evaluating the open-vocabulary performance on COCO is meaningless.
5. The number of decimal places is inconsistent (Line94, Table 1, Table 2).





[1] Exploring Plain Vision Transformer Backbones for Object Detection. In ECCV 2022.

[2] CLIPSelf: Vision Transformer Distills Itself for Open-Vocabulary Dense Prediction. In ICLR 2024.

[3] Grounding DINO: Marrying DINO with Grounded Pre-Training for Open-Set Object Detection. In ECCV 2024.

[4] YOLO-World: Real-Time Open-Vocabulary Object Detection. In CVPR 2024.

[5] Grounded Language-Image Pre-training. In CVPR 2022.

[6] Coarse-to-Fine Vision-Language Pre-training with Fusion in the Backbone. In NeurIPS 2022.

**Questions:**

Please see the weaknesses above.

---

> ### Author Response · Authors · 2025-11-24
>
> Thanks for the detailed feedback and for acknowledging the strengths of our paper, including the clear visualizations, thorough ablation studies, and overall clarity of the writing.
> We would like to address the reviewer's weaknesses as below:
>
> > The paper proposed three independent innovations (Visual-Language Pyramid Upscale Block, Visual-Language Region Proposal Network, and Mini-Batch Image Contrastive Loss) without a unified motivation, making the paper A + B + C.
>
> Our three contributions (VL-PUB, SigRPN, and $L_{ICL}$) are unified under a single goal: effectively adapt pre-aligned, single-scale VLMs (e.g., CLIP) for multi-scale open-vocabulary object detection (OVD):
> 1) VL-PUB and $L_{ICL}$: These modules solve the structural incompatibility between single-scale VLM encoders and multi-scale detection frameworks. VL-PUB constructs the required feature pyramid, while $L_{ICL}$ simultaneously preserves the crucial coarse-grained, image-level alignment from CLIP during the adaptation.
> 2) SigRPN: To the best of our knowledge, it is the first sigmoid-based contrastive RPN that uses semantic information to distinguish "any object" from "background," which is critical for robust generalization to novel categories in OVD.
>
> > Using a multi-scale feature pyramid is a common practice in object detection. As ViT-based CLIP produces single-scale features, we can use ViTDet [1,2] to produce a feature pyramid.
>
> 1) Simply adopting a vision-only architecture like ViTDet for the backbone would discard the invaluable, pre-aligned visual-language latent space of the VLM (e.g., CLIP). Our VL-PUB is novel because it is designed to simultaneously (1) construct the multi-scale feature pyramid and (2) fuse it with text features during the upscaling process. This ensures the fine semantic alignment is preserved and propagated to high-resolution features, which is critical for OVD.
> 2) We also found that naively applying the original CLIP contrastive loss to this new framework causes it to dominate the training process and degrade detection performance (as discussed in the Appendix). $L_{ICL}$ is a specific contribution designed to stabilize this delicate balance, which is a crucial detail for integrating VLMs into OVD frameworks.
>
> > Authors should compare with related works [3,4,5,6]. I don't think the challenges mentioned from Line 164 to Line 167 are a big problem.
>
> 1) We confirm that our paper already compares VLDet against [3, 4, 5] in the paper. To specifically address FIBER [6], we provide the zero-shot comparison below:
> | Method | pre-train data |COCO2017 (zero-shot AP) |
> | --- | --- | --- |
> | FIBER-B | COCO+CC3M+SBU Captions+Visual Genome +Objects365 |49.3  |
> | VLDet-B| Objects365| 43.69|
>
> While FIBER achieves a higher zero-shot $AP$, it requires a five-dataset aggregation for pre-training, even including COCO (not strictly zero-shot). VLDet-B achieves competitive performance (43.69 AP) using a single Objects365 dataset, highlighting our architecture's superior data efficiency.
> We will include the detailed results in the final version.
>
> 2) Sorry for the confusion. We did not mean to imply that other methods cannot achieve high performance. Rather, the complex pipelines of prior works often introduce training complexity and reduced reproducibility. Our design is a neat and efficient training process, enhancing both efficiency and reproducibility.
>
> > The experiments in Table 1 are extremely unfair as the proposed method uses much more pretraining data and a significantly larger backbone (ViT-Large). Authors should include other strong baselines for comparison, including but not limited to [2, 3]. Further, objects365 contains all 80 classes in COCO. Evaluating the open-vocabulary performance on COCO is meaningless.
>
> 1) In fact, our approach is more data-efficient than many baselines. We pre-train solely on Objects365. Many SOTA methods, including YOLO-World, use objects365 in addition to other large-scale image-caption datasets (like CC3M, GoldG) that require complex pseudo-labeling.
> 2) We report results for both ViT-B and ViT-L to ensure fair comparison. Our VLDet-B model (with ViT-B) already outperforms comparable baselines. We include additional comparison with [2] (Both with ViT-L):
> | Method |COCO2017 (AP_Novel) |
> | --- | --- |
> | CLIPSelf-L | 44.3|
> | VLDet-L| 58.72|
> 3) The OVD definition of “novel” is based on the downstream fine-tuning stage, not on the pretraining dataset.
> We follow the standard split: we fine-tune our model only on the 48 base classes of COCO. The evaluation then measures the performance on 17 "novel" classes that it did not see during this fine-tuning stage.
> Objects365 or similar datasets containing COCO objects (e.g., CC3M) are widely used for pre-training, like YOLO-WORLD (O365+CC3M), RegionCLIP (CC3M).
>
> > The number of decimal places is inconsistent (Line94, Table 1, Table 2).
>
> Thanks. We will round all results in our tables to one decimal place in the revised version to improve readability.

---

> ### Comment · Reviewer_J8mg · 2025-11-26
> **Response to the comment**
>
> Thanks authors for providing the rebuttal. However, VLDet-B is trained with a large pretraining dataset and a strong CLIP ViT backbone, and authors compare it with other models with R50 and much fewer pretraining data.
>
> Under a fair setting, Grounding DINO with swin-t (much smaller backbone) backbone and pretraining on the same Objects365 achieves 46.7 zero-shot AP and 56.9 fine-tuned AP on COCO, which are higher than 43.69 AP and 54.14 AP. Note that Grounding DINO do not initialize from a CLIP model.
>
> Further, Vision-language deep fusion is also a common practice in open-vocabulary object detection. Authors should compare the proposed VLFuse with related works in terms of the fusion method, even the performance of VLDet is lower.

---

> > ### Author Response · Authors · 2025-11-26
> >
> > We sincerely appreciate your thoughtful follow-up comments and the opportunity to clarify our work's positioning and contributions.
> > We address your concerns regarding the fair setting, domain focus, and comparison of fusion methods below:
> >
> > **1. Clarification of VLDet’s Focus: OVD vs. ZOD**
> >
> > There are two distinct domains in the literature: Open-Vocabulary Object Detection (OVD), which requires fine-tuning on base classes, and Zero-Shot Object Detection (ZOD), which focuses purely on zero-shot transfer from large pre-training.
> >
> > Our work is fundamentally focused on the **OVD setting**. Our framework, particularly the novel SigRPN module, is tailored for this domain. By fine-tuning on base classes, the SigRPN learns the semantic information necessary to differentiate background from a "general object" using visual-language alignment. This process is essential for robust generalization to novel categories during the evaluation phase, which is the definition of OVD. While we include ZOD comparisons to show VLDet's strong transferability, ZOD performance (e.g., Grounding DINO) is not our primary objective.
> >
> > **2. The "Fairness" of Backbone and Architecture**
> >
> > The concern regarding the comparison of our ViT-L backbone (VLDet) against R50-based baselines (eg. ViLD, RegionCLIP, OV-DQUO) and Swin-T (Grounding DINO) is valid, but it underscores one of our main contributions:
> >
> > - The OVD baselines like ViLD, RegionCLIP or recent OV-DQUO (AAAI 2025) had to rely on ResNet backbones or complex distillation techniques because the single-scale nature of CLIP's ViT made direct adaptation to multi-scale feature pyramids (a necessity for accurate detection) a major challenge. Our work's key innovation is the VL-Pyramid Upscale Block (VL-PUB), which solves this long-standing problem. Using the ViT-B/ViT-L backbone is thus a demonstration of the success of our novel adaptation framework, not an arbitrary advantage.
> > - While our ViT-L backbone is stronger than the Swin-T used by Grounding DINO, Grounding DINO's detector architecture (based on DETR/Transformer) is significantly more powerful than our R-CNN framework so that it can provide better zero-shot/fine-tuned AP. (Grounding DINO targets ZOD domain, which is distinct from our OVD focus.)
> > - Our adoption of ViT rather than Swin is deliberate, as ViT is the standard backbone for SOTA VLMs (e.g., CLIP, SigLIP). This is another main contribution: our unified framework can directly adapt any single-scale VLM ViT into an OVD detector. This strategy offers significant efficiency by avoiding the increasingly expensive VLM pre-training and resource-intensive distillation stages, providing a more direct route to robust OVD performance.
> >
> > **3. Clarification on Pre-training Data Scale (Data Efficiency)**
> >
> > We would like to correct the impression that VLDet is trained with a "large pretraining dataset." In fact, VLDet is highly data-efficient compared to leading OVD baselines. We compare VLDet against OVD baselines that also adopt a pre-training stage (excluding knowledge distillation-based methods for fair comparison):
> >
> > | Method | Pre-train Data | Data Size |
> > |:-----|:---:|-----:|
> > | RegionCLIP | CC3M | 3M |
> > | YOLO-World | Objects365+GoldG+CC3M | 5.5M |
> > | VLDet (Ours) | Objects365 | 1.7M |
> >
> > As shown, VLDet uses fewer samples (≈**1.7 Million**) than baselines. Our performance, achieved with a relatively smaller, detection-focused dataset, further highlights the efficiency and effectiveness of our architectural adaptation and multi-level alignment strategy. For ZOD baselines, the strongest models typically use > 24M samples (CC3M, O365, GoldG, etc.).
> >
> > **4. Comparison of VLFuse with Related Works**
> >
> > We agree that a discussion of our VLFuse module is necessary to complete the comparison. We include the detailed comparison in **Section 4.2** and **Table 2** (added FIBER [6] as suggested, in the revised version).
> >
> > Our VLFuse is a lightweight fusion mechanism, in contrast to the highly complex, token-level deep fusion methods employed by leading ZOD detectors, such as the heavy fusion block in GLIP (VLDyHead) or the deeper, transformer-based fusion/query interaction throughout the detector layers in Grounding DINO.
> >
> > While the fundamental strength of the DETR architecture allows Grounding DINO to achieve the highest ZOD/fine-tuned AP, we observe that VLDet achieves comparable Zero-Shot Detection (ZOD) performance to GLIP (which utilizes a more similar detection framework). This result strongly demonstrates the efficacy of VLFuse when paired with our multi-level fine-grained alignment losses. The success on ZOD validates the robustness of our vision-language alignment, even though our architectural design choices were primarily optimized for the OVD domain.
> >
> > We believe these clarifications firmly establish VLDet's novel contributions in adapting powerful ViT backbones for OVD with impressive data efficiency and a strong focus on generalization. If you have any more comments, please let us know.

---

### Official Review · Reviewer_29mf · 2025-11-01

**Soundness:** 3
**Presentation:** 3
**Contribution:** 3
**Rating:** 8
**Confidence:** 4

**Summary:**

The paper proposes VLDet, a two-stage OVD framework that (i) adapts CLIP backbones into a multi-scale detector via a VL-Pyramid Upscale Block (VL-PUB) and (ii) introduces a Sigmoid-based RPN (SigRPN) trained with anchor–text binary alignment, plus mini-batch image–caption and region–text contrastive losses for multi-level alignment. On COCO and LVIS, the method reports large gains on novel categories (e.g., 58.72 AP novel on COCO; 24.83 AP novel on LVIS) after pretraining only on Objects365 with generated captions. Ablations cover freezing strategies, multi-scale vs single-scale, caption usage, and loss components.

**Strengths:**

1. The model achieves state-of-the-art novel-class average precision on COCO and LVIS benchmarks, given its specified pretraining data. A series of well-designed ablations—including those on captions, language-image contrastive learning, visual-language fusion, and multi-scale features—credibly demonstrate the source of its performance gains.

2. The proposed SigRPN module is an elegant design. By framing region proposal as a binary visual-language alignment task for foreground and background separation, it provides a clear and effective method for improving proposal quality for unseen object categories.

3. The paper provides sufficient implementation details for reproducibility, reporting specific choices such as batch sizes, components kept frozen during training, and comparative results against single-scale baselines.

**Weaknesses:**

1. The substantial requirement of 64 A100 GPUs for pretraining will likely hinder widespread adoption. Furthermore, the paper does not profile the inference cost, making it difficult to assess its practical efficiency compared to prior two-stage open-vocabulary detectors.

2. While the method of using a foundation vision-language model to generate captions for the Objects365 dataset is described, the exact model, prompts, and data cleaning procedures are not fully specified. This lack of detail complicates strict reproducibility and fair comparison with future work.

3. The comparison to some baselines, such as recent grounding-detector variants, is complicated by differences in their pretraining data. Although the paper acknowledges this, a stricter, like-for-like evaluation protocol would strengthen its claims.

**Questions:**

1. Please specify which captioning model and prompts were used for the Objects365 dataset, and will this generated caption set be released to ensure reproducibility?

2. Is it possible to report the inference latency and FLOPs for the VLDet-B/L models, both with and without SigRPN, and compare them to common open-vocabulary detection baselines?

3. How sensitive are the final results to the freezing of the vision-to-language module, and to the choice of the underlying CLIP model variant, when evaluated under a consistent computational budget?

---

> ### Author Response · Authors · 2025-11-24
>
> We are very grateful to the reviewer for their positive assessment and insightful feedback. We are especially encouraged that the reviewer found our SigRPN module to be an "elegant design" and appreciated our "well-designed ablations" for credibly demonstrating the source of our performance gains.
> Below, we address the concerns point by point:
>
> > The substantial requirement of 64 A100 GPUs for pretraining will likely hinder widespread adoption.
> > Is it possible to report the inference latency and FLOPs for the VLDet-B/L models, both with and without SigRPN, and compare them to common open-vocabulary detection baselines?
>
> We thank the reviewer for raising this important point on practical efficiency.
> 1) The 64 A100 GPU count was used to accelerate the one-time pre-training on the Objects365 dataset to reduce wall-clock time. We have verified that the identical accuracy can be achieved on a more modest 8–16 GPUs, albeit with a longer training time.
> 2) Pretraining requires ~1100 GPU-hours for VLDet-B and ~2600 GPU-hours for VLDet-L, which is substantially lower than methods relying on millions of pseudo-labeled region annotations (e.g., GLIP, and Grounding DINO).
> 3) As stated in the paper, the introduced layers for VLDet require only 221 GFLOPs. This is significantly more efficient than the 407 GFLOPs introduced by the VLDyHead in GLIP. For inference cost, we profiled VLDet-B on an A100:
> | Model  | GFLOPs | FPS |
> | :------- | :------: | :------: |
> |VLDet-B (w/o SigRPN) |  481 | 18|
> | VLDet-B (w/ SigRPN)   | 561  | 14|
> | GLIPv2-T   | 674  | 11 |
> We will add more details to the revised version.
>
> > While the method of using a foundation vision-language model to generate captions for the Objects365 dataset is described, the exact model, prompts, and data cleaning procedures are not fully specified.
> > Please specify which captioning model and prompts were used for the Objects365 dataset, and will this generated caption set be released to ensure reproducibility?
>
> 1) The captions for Objects365 were generated using the BLIP-2 model.
> 2) We used a standard, straightforward prompt for the generation: “Describe this image in one sentence.”
> 3) The entire Objects365 dataset was used, and no further data cleaning was applied to the generated captions.
>
> We will add these specific details to the Implementation Details section of the revised manuscript.
> Furthermore, to ensure full reproducibility and to benefit future research, we will publicly release the complete generated Objects365 caption dataset alongside our final code and pre-trained model weights. (We have already submitted the application for public data release to our institute.)
>
> > The comparison to some baselines, such as recent grounding-detector variants, is complicated by differences in their pretraining data. Although the paper acknowledges this, a stricter, like-for-like evaluation protocol would strengthen its claims.
>
> While it is a common practice in OVD literature to compare against SOTA baselines that leverage different pre-training datasets, so long as the data sources are clearly listed, we agree with the reviewer that a stricter, like-for-like evaluation provides valuable insight and strengthens our claims.
> To address this point directly, we conducted a new experiment by re-training the YOLO-WORLD-M baseline using only the Objects365 dataset, which is identical to the pre-training data of our VLDet-B.
> The results are below:
> | Method | Pre-train Data | LVIS ($AP_r$) |
> | :------- | :------: | :------: |
> | YOLO-WORLD-M | Objects365| 13.7|
> | VLDet-B   | Objects365 | 16.93  |
> Our VLDet-B model outperforms YOLO-WORLD-M by a significant margin of +3.23 $AP_r$ when both models are trained on the exact same data. As expected, the performance of this re-trained YOLO-WORLD-M (13.7 $AP_r$) is lower than its officially reported score, which benefited from additional large-scale datasets.
>
> > How sensitive are the final results to the freezing of the vision-to-language module, and to the choice of the underlying CLIP model variant, when evaluated under a consistent computational budget?
>
> 1) Sensitivity to Freezing: This is a critical component of our fine-tuning strategy, which we analyze in Section 4.3 and Table 5. We found that freezing the V2L (visual-to-language) layers is essential. If no layers are frozen, the model severely overfits to the base classes, causing a catastrophic drop in novel-category performance (e.g., $AP_r$ on LVIS degrades from 24.83 to 14.77)
> 2) To more directly address the question of different VLM types under a consistent computational budget, we conducted a new experiment comparing the standard CLIP (ViT-B) to the SigLIP (ViT-B).
>  | Vision backbone  | COCO ($AP_{Novel}$) |
> | :------- | :------: |
> | CLIP |  54.2|
> | SigLIP   | 56.8  |
> The results show a clear benefit from using the stronger VLM, which indicates that our framework is robust and can readily leverage advances in VLM pre-training to achieve further performance gains.

---

### Meta-Review · Area_Chair_LZup · 2025-12-28

**Summary:**

The paper proposes VLDet, a two-stage open-vocabulary detector adapting CLIP. It proposes three improvements to tackle the open-vocabulary object detection task. First, the Visual-Language Pyramid Upscale Block extracts multi-level feature pyramid from the single-scale image feature to detect objects at various scales. Second, the Visual-Language Region Proposal Network fuses image features with text embeddings and uses an Anchor-Text Binary Alignment Loss for classification. Third, a Mini-Batch Image Contrastive Loss is used to transfer image-level knowledge.
However, concerns about fairness of comparisons, unclear unified motivation, and incomplete methodological details limit confidence in its overall contribution.
Considering the reviewers’ concerns, we regret that the paper cannot be recommended for acceptance at this time. The authors are encouraged to consider the reviewers’ comments when revising the paper for submission elsewhere.

**Reviewer Concerns:**

Reviewers highlight (1) lack of unified motivation across components, (2) Using a multi-scale feature pyramid is a common practice in object detection, (3) Vision-language deep fusion is also a common practice, (4) concerns in comparison fairness.

**Reviewer Scores:**

Scores are mixed, ranging from acceptance to rejection. While some reviewers rate soundness and presentation highly, others judge contribution and experimental fairness as only fair.

---

### Decision · Program_Chairs · 2026-01-26

Reject